# Real-Time Assessment of Intracellular Metabolites in Single Cells through RNA-Based Sensors

**DOI:** 10.3390/biom13050765

**Published:** 2023-04-28

**Authors:** Alvaro Darío Ortega

**Affiliations:** Department of Cell Biology, Faculty of Biological Sciences, Complutense University of Madrid, 28040 Madrid, Spain; ad.ortega@ucm.es; Tel.: +34-913-944-982

**Keywords:** biosensor, RNA-based sensor, aptamer, metabolism, metabolite, fructose-1,6-bisphosphate, metabolic flux, ribozyme, riboswitch, spinach

## Abstract

Quantification of the concentration of particular cellular metabolites reports on the actual utilization of metabolic pathways in physiological and pathological conditions. Metabolite concentration also constitutes the readout for screening cell factories in metabolic engineering. However, there are no direct approaches that allow for real-time assessment of the levels of intracellular metabolites in single cells. In recent years, the modular architecture of natural bacterial RNA riboswitches has inspired the design of genetically encoded synthetic RNA devices that convert the intracellular concentration of a metabolite into a quantitative fluorescent signal. These so-called RNA-based sensors are composed of a metabolite-binding RNA aptamer as the sensor domain, connected through an actuator segment to a signal-generating reporter domain. However, at present, the variety of available RNA-based sensors for intracellular metabolites is still very limited. Here, we go through natural mechanisms for metabolite sensing and regulation in cells across all kingdoms, focusing on those mediated by riboswitches. We review the design principles underlying currently developed RNA-based sensors and discuss the challenges that hindered the development of novel sensors and recent strategies to address them. We finish by introducing the current and potential applicability of synthetic RNA-based sensors for intracellular metabolites.

## 1. Introduction

Cell metabolism, the set of enzyme-catalyzed chemical reactions occurring in the cell leading to energy production and biomass formation, works as a coordinated system that results from the interaction of multiple metabolic pathways, where metabolites act as interconnecting nodes among these pathways. Cells regulate metabolic fluxes to adapt their metabolic phenotype to environmental conditions [1,2,3,4]. Thus, the assessment of metabolic fluxes directly reports on the metabolic phenotype. Isotope tracing is an end-point approach performed at the cell population level, where an average response is measured; its implementation requires trained personnel and expensive equipment [5]. Therefore, isotope tracing is not a suitable approach to measure metabolic fluxes in real-time and/or at the single-cell level and is unaffordable to implement in most cell biology laboratories.

In recent years, synthetic biologists have developed novel cellular devices that make it possible to assess the intracellular concentration of a small molecule by quantifying the fluorescence emitted by individual cells in a population. These sensors consist of a macromolecule (protein or RNA) expressed by the cell that has a domain that interacts specifically with the small molecule (e.g., a metabolite) (the *sensor domain*) and a different part that allows converting this interaction into a quantifiable response (e.g., fluorescence) (the *actuator*) [6,7]. In this work, we describe molecular mechanisms employed by cells across all kingdoms to sense intracellular metabolites and regulate cell responses, focusing on those mediated by cellular RNA molecules. We review the development of synthetic RNA devices designed to serve as biosensors of intracellular metabolites and analyze the major technical issues and solutions in this approach. We finish by pointing at current and future applications of synthetic RNA-based sensors of metabolites. 

## 2. Cells Sense Intracellular Metabolites and Transduce This Information into Cellular Responses

Cells adapt to dynamic environmental conditions. Upon changes in nutrient availability, eukaryotic and prokaryotic cells reshape their molecular machinery to optimize the uptake and metabolism of particular carbon sources. Membrane-bound protein sensors can mediate the detection of the extracellular concentration of specific nutrients. At least six different types of membrane receptors are involved in chemotaxis in *E. coli* in response to a broad spectrum of extracellular nutrients, including amino acids, sugars, and nucleotides [8]. By coupling motility to nutrient-sensing, bacteria are more suited for nutrient scavenging in limiting conditions. In yeast, membrane receptors Snf3 and Rgt2 sense low and high glucose concentrations, respectively, and initiate intracellular signal transduction cascades that lead to the induction of membrane transporters with different affinities [9], thereby adjusting glucose uptake to its actual extracellular concentration. However, the occurrence in mammals of homologous systems that sense extracellular levels of glucose or amino acids through membrane-bound receptors could not be proven so far. 

Cells can take up multiple nutrient species, which are ultimately utilized through a reduced set of metabolic pathways. Asthis diverse set of carbon, nitrogen, and sulfur sources is funneled intracellularly into a few metabolic intermediates, the generation of dedicated specific membrane-bound sensors might be an unjustified evolutionary effort. In fact, evidence accumulated over decades indicates that, instead, cells sense nutrient availability through intracellular sensors [3,4,10,11,12]. Intracellular sensors bind specific metabolites or second messengers, which then generate a downstream effect that allows cells to adapt to changes in nutrient availability. For instance, α-proteobacteria sense nitrogen scarcity by monitoring the intracellular concentration of glutamine and 2-oxoglutarate. The balance between these two metabolites ultimately controls the activity of glutamine synthase, the enzyme that catalyzes the first step of ammonium assimilation [13]. In eukaryotes, nitrogen availability is also sensed by intracellular molecules. The TOR/mTOR and GCN2-GCN4/ATF4 pathways, which are conserved from yeast to mammalian cells, sense amino acid availability and, in turn, control cell metabolism and mediate growth decisions and general stress response [14]. Interestingly, the intracellular signal that relays amino acid scarcity through GCN2 in eukaryotes (uncharged tRNAs) also triggers a general stress response that inhibits protein translation and stops cell growth in bacteria [15]. Therefore, intracellular nutrient sensing and signaling could converge into a few specific metabolites across evolution. 

One such example is the glycolytic intermediate fructose-1,6-bisphosphate (FBP). FBP levels change in cells grown under different nutritional regimes and respond to dynamic perturbations of glycolysis across a spectrum of bacterial and yeast species [4,10]. FBP not only correlates with glycolytic flux but also transduces it into cellular responses, and then it is often referred to as a *flux-signaling metabolite* [3,16]. Evidence of glycolytic flux sensing through FBP has been found in bacteria, yeast, and mammalian cells. For instance, in Gram-positive bacteria, carbon catabolite repression (CCR) is mediated by the CcpA transcription factor [17]. When glucose is available, FBP stabilizes CcpA binding to DNA, which represses genes involved in the uptake and metabolism of alternative carbon sources [18,19]. In addition, FBP induces the expression of glycolytic enzymes by prompting the dissociation of the CggR repressor from the promoter of the *gapA* operon [20,21,22]. In yeast, high FBP concentration favors fermentative over oxidative metabolism by activating the H-Ras homolog Ras2 [23,24]. Finally, in human cells, FBP inhibits the energy sensor AMP-dependent protein kinase (AMPK) through an AMP/ADP-independent mechanism [25]. Thus, as a glycolytic flux-signaling metabolite, FBP serves as an intracellular signal of glucose availability in all kingdoms of life. 

Overall, cells from bacteria to humans utilize specific intracellular metabolites as cues to probe nutrient availability and metabolic fluxes and transduce this information to generate adaptive cellular responses. Therefore, the development of sensors suitable to assess the intracellular concentration of such metabolites could serve as tools to monitor metabolic fluxes in cells. 

## 3. Prokaryotic Cells Utilize RNA to Sense Intracellular Metabolites

An important endeavor in synthetic biology is to develop nature-inspired systems with programmable architecture and function, such as engineered biosensors. These biosensors are molecular devices that respond to small molecules generating a quantifiable signal or specific functionality. Ideally, they should display a modular architecture, with at least a *sensing* moiety that binds the small molecules and a *reporter* moiety that triggers the specific response. This way, different modules could be combined to build sensors with different specificities using the same basic design or sensors that generate different readouts in response to the same small molecule [26]. In the previous section, I have described examples where intracellular metabolites (or second messengers) bind protein sensors. The conformational changes induced upon binding modulate protein activity, which eventually results in a cellular effect. However, the molecular mechanisms that transduce metabolite-binding into a cellular effect through protein sensors are diverse. Thus, the design of natural systems involving protein sensors cannot be easily replicated and systematized to build engineered sensors.

Bacteria have a broadly-represented molecular sensing mechanism based on RNA [27]. These natural prokaryotic sensors named riboswitches are RNA segments that occur in two conformations, the transition between which is favored by binding to a metabolite. Riboswitches are mostly found in the untranslated regions at the 5′ (5′-UTR) of mRNAs, and each of the two conformations has a different impact on downstream gene expression: the *ON* conformation allows gene expression while the *OFF* conformation halts it. Riboswitches are made of two structurally and functionally differentiated modules: a sensing domain (or *aptamer*) that binds the metabolite, and an *expression platform*, which modulates gene expression as a result of the structural changes induced upon ligand binding [28]. The riboswitch-regulated downstream gene is functionally associated with the metabolite ligand. For example, in the cyanobacteria *Synechocystis*, glutamine binds a riboswitch that activates the expression of an inhibitory factor of the glutamine synthase (GS) [29]. This riboswitch belongs to a broader class, named *glnA* riboswitch, frequently found in the 5′-UTR of genes involved in nitrogen metabolism in cyanobacteria and marine metagenomic sequences, including the GS, glutamate synthase, and ammonium transporters. This functional association indicates that glutamine riboswitch is a central gene regulatory element of nitrogen assimilation that respond to intracellular glutamine levels in cyanobacteria [30].

More than 55 classes of riboswitches, with different ligand binding specificities and aptamer structures, have been experimentally validated thus far. They respond to a broad spectrum of intracellular ligands, including enzyme cofactors (e.g., derivatives of vitamin B, S-adenosylmethionine, FMN, NAD), nucleotide derivatives bases (e.g., guanine, adenine, xanthine), nucleotide-based signaling molecules (e.g., cyclic di-GMP, cyclic di-AMP, guanosine tetraphosphate ppGpp, ZTP), amino acids (e.g., lysine, glycine, glutamine), elemental atoms (e.g., Mn^2+^, Mg^2+^, Li^+^, Na^+^) and other small molecules [31]. The diversity of intracellular molecules sensed by riboswitches, as well as the fact these ligands carry out crucial functions in fundamental metabolic processes in organisms across all kingdoms of life, endorse the success of riboswitches as natural sensors of metabolism. Interestingly, the pervasiveness in modern bacteria of nucleotide derivatives as riboswitch-binding and signaling molecules might represent a vestige of an ancient RNA world. Consistent with this hypothesis, it is possible that ligand-binding domains in today’s riboswitches could have been *recycled* from a primordial RNA-based catalytic machinery in the RNA world that transformed or synthesized these small molecules in metabolism [32,33]. 

Although riboswitches can bind diverse intracellular ligands, the response is generated through a reduced set of molecular mechanisms. In most cases, this regulatory activity is directed at transcription or translation by ligand-dependent stabilization of one of the two mutually exclusive RNA conformations. In the case of transcription, ligand binding favors the formation of a stable double-stranded RNA stem structure followed by a polyuridine tract, which serves as a transcription terminator. The formation of this terminator promotes a premature termination of transcription of the downstream gene. Under low ligand concentrations, the riboswitch folds as an anti-terminator structure, which allows for a transcriptional read-through. In the case of translation, the conformational switch induced by the ligand influences the accessibility of the ribosome-binding site (RBS) and the initiation codon. In the OFF conformation, these sequence elements become occluded within a stem, which halts the initiation of translation [34]. The *glmS* riboswitch represents a widespread class that uses a singular mechanism. It is a sequence-specific self-cleaving ribozyme whose endonucleolytic activity depends on *N*-acetyl-glucosamine-6-phosphate (GlcN6P) [35,36]. Interestingly, structural and biochemical data suggest that GlcN6P does not induce a conformational switch but participates directly in RNA cleavage activity [28,37]. Additional but less frequent mechanisms that rely on regulatory proteins have also been described [38]. 

Conserved sensing domains can occur in combination with different types of expression platforms. For instance, thiamin pyrophosphate (TPP)-binding riboswitch (Thi-box) is widely distributed in the three kingdoms of life. In *Escherichia coli*, the binding of TPP prevents translation initiation by sequestering the RBS and the start codon, while in *Bacillus subtilis,* it leads to transcription attenuation [39,40]. In eukaryotes, the riboswitch occurs within introns or 3′-UTRs, where TPP binding makes an occluded splice site accessible to the splicing machinery. Isoforms derived from this alternative splicing event either lead to the occurrence of premature stop codons in the mature mRNA or to the destabilization of the transcript, depending on the species [34]. Thus, similar ligand-induced conformational changes in conserved sensing RNA domains can trigger different regulatory activities depending on the adjacent expression platform. This observation indicates that riboswitches are made of functional modules. In fact, this natural modular architecture has been implemented in synthetic RNA devices, where different expression platforms could be coupled to a variety of natural and synthetic aptamers [41,42].

Overall, riboswitches are naturally-evolved sensors for intracellular metabolites that serve to monitor intracellular metabolism. They show a modular architecture. This means that a defined set of RNA-sensing domains with different specificities or aptamers can be combined with a few basic gene regulatory modules to generate the cellular response. These properties make riboswitches a suitable model to simulate for engineering synthetic RNA-based biosensors for metabolites. 

## 4. Synthetic RNA-Based Sensors for Intracellular Metabolites

A synthetic RNA-based sensor is a genetically-encoded RNA molecule consisting of a metabolite-binding aptamer (*sensor* domain) and a transducer region that converts the interaction of the metabolite into a quantifiable response (the *actuator* domain) (Figure 1A). The first demonstration of a synthetic RNA-based sensor was an allosteric ribozyme developed by Tang and Breaker in 1997 [43]. By using a rational design, an ATP-binding aptamer (sensor domain) [44] was joined to a hammerhead ribozyme (actuator domain) through different linker sequences. The generated constructs were then screened for an ATP-driven inhibition of the self-cleaving activity of the ribozyme in vitro [43]. Testing different linker regions is a necessary step to identify the construct that more efficiently transduces metabolite-binding into the output signal [45]. Although this primordial RNA device was not genetically encoded, it was the first proof-of-principle of an engineered genetic device de novo whose activity could be controlled by a small molecule. 

Since then, a plethora of RNA devices that control gene expression in a ligand-dependent fashion have been developed. Such systems utilized a reduced set of sensing aptamers to prove valid different mechanisms of actuation. These synthetic RNA-device prototypes usually included the aptamer of theophylline [46], tetracyclin [47], or thiamin pyrophosphate (TPP) [39] coupled to an actuation domain that allowed for allosteric control of the functionality of translational machinery [48,49], translation initiation [47,50,51,52], mRNA stability [53,54,55], splicing [56] and microRNA (miRNA) maturation [57]. Some devices are exclusively functional in either bacteria or eukaryotes because the functionality of the actuator domain relies on kingdom-specific gene expression control mechanisms. For instance, the mechanisms that regulate the initiation of protein translation are different in bacteria and eukaryotes, while splicing or miRNA-based regulation occurs only in eukaryotes. Ideally, programmable RNA-based gene-regulatory devices should rely on a *portable* mechanism of actuation, that is, a transduction mechanism that is independent of cell-specific machinery [54]. 

Fast-cleaving variants of the hammerhead ribozyme (HHRz) have been widely utilized as actuator domains in RNA devices. These HHRz-based RNA devices are functional across different organisms, where they showed to respond to diverse ligands [50,52,58,59,60], generating varied responses, including the regulation of complex cell phenotypes [53,61]. The architecture of such devices consists of a small molecule-binding RNA aptamer grafted into the HHRz in place of one of its stems II or III (Figure 1B). To implement the allosteric ribozyme (or *aptazyme*), linker sequences are rationally designed to relay ligand-binding into a secondary structure change [54] or randomized and screened for ligand-dependent impact on self-cleaving activity [58]. An alternative approach aims to select synthetic allosteric ribozymes where ligand binding interferes with or contributes to stabilizing the tertiary interactions between the two loops of the HHRz (Figure 1B). In this case, aptamer-coupled ribozyme libraries are constructed by varying the sequence of the loop opposite to the stem-loop where the sensing domain is grafted [59,60,62].
Figure 1RNA-based sensor frameworks implemented to detect intracellular metabolites. (**A**) Basic design of a sensor. (**B**) Aptazyme-based sensor. Hammerhead (HH) endonucleolytic activity depends on the binding of the metabolite (Orange; FBP, fructose-1,6-bisphosphate). Active ribozyme destabilizes the mRNA [62]. (**C**–**F**) Light-up sensors. Folding of a fluorogenic aptamer (Grey; Spinach, Broccoli) depends on the binding of the metabolite to the sensing aptamer (blue; ADP, TPP, SAM aptamers). When folded, the fluorogenic aptamer can bind a fluorophore (DFHBI, DFHBI-1T), generating fluorescence. Three-way junction motif: 3WJ. DFHBI-1T: 1T. (**C**) [63]. (**D**) [64]. (**E**) [65]. (**F**) [66].
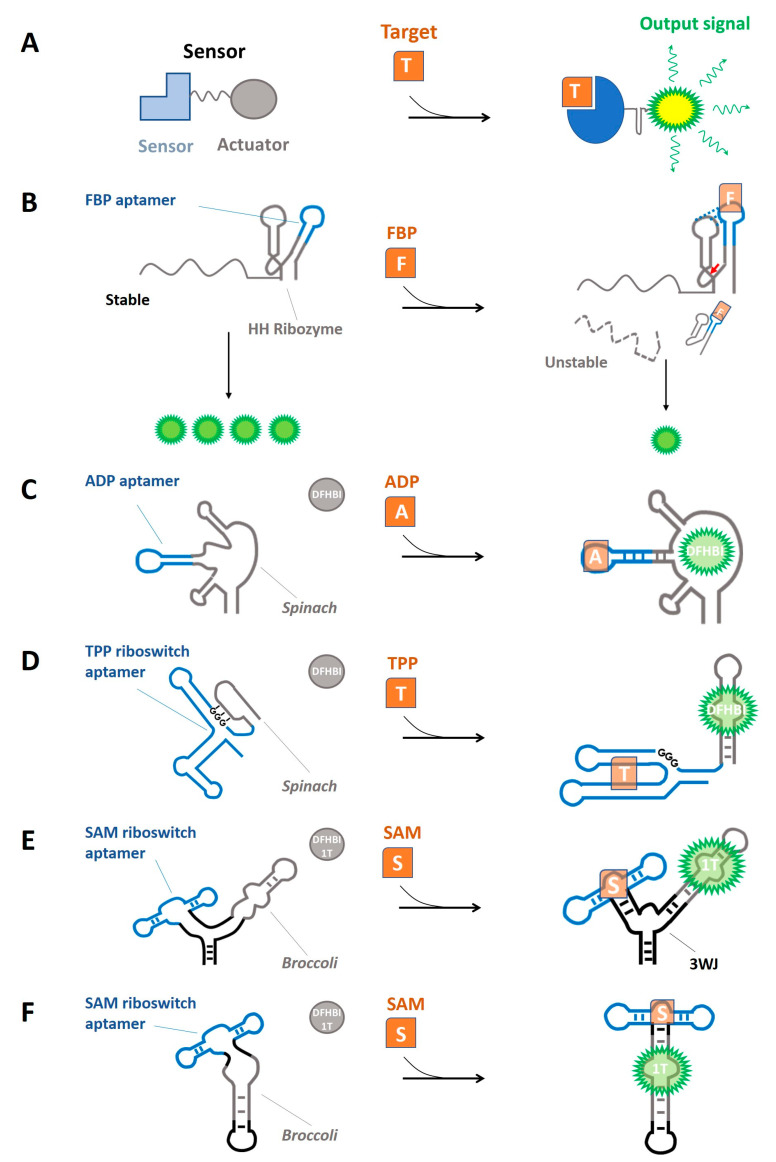



RNA devices based on the allosteric HHRz framework have been implemented as metabolite sensors for the vitamin B derivative TPP, the glycolytic metabolite fructose-1,6-bisphosphate (FBP), the product of catabolism of purines hypoxhanthine, and the bacterial second messenger cyclic diguanylate monophosphate (c-di-GMP) (Table 1). TPP biosensor was built by inserting the metabolite-sensing domain of TPP riboswitch into the 5′-UTR of a GFP expression cassette in *E. coli*, such that TPP-dependent activation of HHRz activity controls the blockade of the RBS [67]. In contrast, in the FBP, hypoxhantine, and c-di-GMP biosensors, the aptazyme is located in the 3′-UTR of a reporter gene and exhibits metabolite-responsive HHRz tertiary structure interactions. Thus, intracellular metabolite concentration-dependent cleavage leads to mRNA destabilization and a concurrent decrease in the abundance of a fluorescent protein (Figure 1B) [60,62].

In addition to HHRz aptazyme sensors, fluorogenic or light-up RNA-based sensors represent a modular, scalable, and portable framework for engineering genetically-encoded biosensors for intracellular metabolites [54,68]. Light-up sensors rely on fluorogenic RNA aptamers, such as *Spinach* [63]. *Spinach* interacts specifically with a non-fluorescent small molecule (3,5-difluoro-4-hydroxybenzylidene imidazolinone, DFHBI) similar to GFP fluorophore, which upon binding to the aptamer, becomes fluorescent [69]. Additional fluorogenic RNA aptamers (with their respective target fluorophores) with different biophysical properties have been developed and implemented into RNA sensors, such as *Broccoli* (DFHBI-1T) [70] and *Corn* (DFHO) [71]. In light-up RNA-based sensors, folding of the fluorogenic aptamer (actuator domain) requires binding of the ligand to a metabolite-binding aptamer (sensor domain), such that ligand-induced stabilization of a structural element allows the recruitment of the fluorophore and fluorescence emission (Figure 1C–F). 

Light-up RNA sensors for intracellular metabolites were initially constructed by joining the sensing aptamer to Spinach through a weak transducer stem that stabilized upon ligand binding (Figure 1C) [63]. The efficacy of the RNA secondary structure changes to transduce a different functional state of the actuator domain (i.e., the *switching activity*) was later improved in a second generation of light-up sensors named *Spinach riboswitches*. In this sensor framework, the actuator domain of natural riboswitches is replaced by Spinach (Figure 1D). This strategy streamlines sensor development because it repurposes a naturally evolved and in vivo functional switching activity to actuate a synthetic device [64]. In addition, a third generation of light-up sensors exhibits much higher sensitivity thanks to improved RNA folding and stability in eukaryotic cells (Figure 1E,F) [65,66,72]. 

An interesting feature of light-up sensors compared to aptazymes is reversibility. Ribozymes catalyze an irreversible endonucleolytic cleavage of reporter mRNA upon a change in ligand concentration, while light-up sensors can be switched on and off over time. These intrinsic properties make aptazyme-based sensors more sensitive to low target concentrations. In contrast, light-up sensors are more suitable for dynamic measurements, where they can serve to monitor oscillations in the concentration of an intracellular metabolite [66,72]. In addition, while light-up sensors can generate solely fluorescence as the output signal, aptazymes can control the expression of any gene. Thus, aptazyme sensors can transduce metabolite concentration into different cell functionalities, such as a selective advantage or a deleterious protein [73]. In any case, both sensor types are compatible and could be combined depending on the application’s needs. Furthermore, the modular architecture of RNA devices allows for combining an aptazyme and a fluorogenic aptamer actuator domains in an individual sensor. For instance, You et al. coupled a TPP-dependent ribozyme to an unfolded form of the fluorogenic aptamer Broccoli. Ligand-dependent endonucleolytic cleavage of the sensor releases Broccoli, which then folds and generates the signal. This design achieves a higher sensor sensitivity because a single target molecule can induce the cleavage of multiple copies of the sensor and thus is suitable for detecting low-abundance metabolites [74].
biomolecules-13-00765-t001_Table 1Table 1RNA-based sensors for intracellular metabolites validated in cells.Target Metabolite ^1^Aptamer ^2^Type of Sensor ^3^Output ^4^Cell TypeReference5-HTSELEXLight-up (Broccoli)GFBacteriaPorter et al., 2017 [75]ADPSELEXLight-up (Spinach)GFBacteriaPaige et al., 2012 [63]c-AMP-GMPNaturalLight-up (Spinach)GFBacteriaKellenberger et al., 2013 [76]c-di-AMPNaturalLight-up (Spinach2)GFBacteriaKellenberger et al., 2015 [77]c-di-GMPNaturalAptazymeRF/BFMammalianXiang et al., 2019 [60]c-di-GMPNaturalLight-up (Spinach)GFBacteriaKellenberger et al., 2013 [76]c-di-GMPNaturalLight-up (Spinach2)GFBacteriaWang et al., 2016 [78]FBPSELEXAptazyme (HHRz)GF/RFYeastOrtega et al., 2021 [62]FBPSELEXLight-up (Baby Spinach)GFMammalianGeraci et al., 2022 [79]L-DOPASELEXLight-up (Broccoli)GFBacteriaPorter et al., 2017 [75](p)ppGppNaturalLight-up (Broccoli)GFBacteriaSun et al., 2021 [80](p)ppGppNaturalBRET (nLuc donor and Pepper acceptor)GFBacteriaMi et al., 2023 [81]SAHNaturalLight-up (cpSPinach2)GFBacteriaSu et al., 2016 [82]SAMNaturalBRET (nLuc donor and Pepper acceptor)GFBacteriaMi et al., 2023 [81]SAMNaturalLight-up (Broccoli)GFMammalianLitke et al., 2019 [66]SAMNaturalLight-up (Broccoli)GFMammalianMoon et al., 2021 [65]SAMNaturalLight-up (Corn)YFMammalianKim et al., 2019 [71]SAMNaturalLight-up (Red Broccoli)RFMammalianLi et al., 2020 [83]SAMNaturalLight-up (Spinach)GFBacteriaPaige et al., 2012 [63]SAMNaturalLight-up (Squash and Broccoli)YF/GFMammalianDey et al., 2022 [72]TPPNaturalAptazymeGFBacteriaWieland et al., 2009 [67]TPPNaturalAptazyme + Light-up (Broccoli)GFBacteriaYou et al. 2019 [74]TPPNaturalLight-up (Spinach)GFBacteriaYou et al., 2015 [64]XanthineSELEXAptazymeRF/BFMammalianXiang et al., 2019 [60]^1 ^Intracellular metabolite (no exogenous compounds). ^2^ Natural: metabolite-sensing domain of a riboswitch; SELEX: in vitro-selected aptamer. ^3^ Only portable actuators (trans-kingdom actuation mechanisms). ^4^ GF: green fluorescence; YF: yellow fluorescence; RF: red fluorescence; YF/GF, RF/BF, GF/RF: ratio of fluorescence intensity. 


## 5. Development of RNA-Based Sensors for Intracellular Metabolites: A Challenging Endeavor

*Scalability* is a paramount design principle of RNA-based sensors because of their modular architecture. It refers to the possibility of generating new sensors by implementing different small molecule-binding aptamers into standardized sensor frameworks, such as those relying on the aptazyme or light-up actuator domains [54]. However, the diversity of available RNA-based sensors for intracellular metabolites available today is strikingly scant. To my knowledge, there are less than 15 RNA-based sensors with different specificities (Table 1). This indicates that although RNA-based sensors are, in principle, scalable, there are still important challenges that halt the implementation of sensors with novel specificities.

Most light-up RNA sensors have been constructed utilizing metabolite-binding aptamers from naturally-occurring riboswitches (Table 1). This has three major advantages. First, the affinity and selectivity of the aptamer have naturally evolved to interact with the target metabolite under physiological conditions. Second, the switching activity of in vivo functional riboswitches is a mechanism of actuation evolved as a gene-regulatory module. Third, publicly available crystal structures of riboswitches, and especially those in different switched states, serve to identify the mechanism of switching and the sequence elements involved [68,84]. This way, ligand binding, and the derived switching activity are already optimized by nature to work within cells. On the other hand, knowing the mechanism of switching smoothens the rational design of the sensor. However, the potential of generating sensors with novel specificities is limited by the availability of identified aptamers in natural riboswitches. It is possible to expand the catalog of available aptamers through computational searches for novel riboswitches. In this regard, comparative genomics and metagenomics in bacteria and archaea aimed to identify novel structured RNA motifs in intergenic regions unveiled a number of new riboswitch classes [85,86], and it has been proposed that most riboswitch classes among the global bacterial population are still to be discovered [31,87]. Novel riboswitches can also be identified by means of experimental approaches. For instance, the binding of a metabolite ligand triggers an RNA conformational change in naturally occurring aptamers, which can be mapped at the transcriptome-wide level in vitro and in vivo [88,89,90].

The most relevant feature contributing to the scalability of engineered RNA-based sensors is the faculty to generate custom RNA-sensing moieties by means of the systematic evolution of ligands by exponential enrichment (SELEX). SELEX is a general iterative approach that allows for aptamer selection from an RNA combinatorial library (or *pool*) based on the capacity to bind a specific metabolite in vitro [91,92]. Despite the enormous potential of applying SELEX to generate aptamers de novo for RNA-based sensors, to my knowledge, only five sensors for intracellular metabolites have been constructed thus far with in vitro selected aptamers: hypoxanthine, FBP, ADP/ATP, 5-hydroxytryptophan (5-HT) and 3,4-dihydroxyphenylalanine (L-DOPA) (Table 1) [44,62,63,75,93,94].

There is a number of issues related to small molecule-aptamer selection and its integration into RNA device frameworks that could halt the construction of custom sensors [92]. In vitro-selected aptamers may not be functional within cells because the physicochemical conditions used for in vitro selection are different from those found in the intracellular niche. These differences might affect, for instance, the affinity and selectivity of the aptamer for the target metabolite. For FBP aptamer selection, both SELEX and the biochemical characterization (determination of the affinity, selectivity, and ligand-induced structural changes) were performed in a buffer in which the pH, the ionic composition, and the concentration of non-target metabolites and macromolecules were adjusted to resemble the conditions of eukaryotic cytoplasm [62,93]. In addition, in vitro-selected aptamers might not fold porperly within cells and/or might exhibit reduced stability. This is the case of Spinach, for example, which requires being embedded within a folding scaffold to yield an optimal signal [95,96]. To circumvent folding and stability issues, aptamers could be selected in vitro embedded within a natural RNA scaffold. Porter et al. identified the three-way junction motif as a common RNA motif in bacterial riboswitches and utilized it as a scaffold to generate the pool for SELEX. They selected aptamers for neurotransmitter precursors 5-HT and L-DOPA, which were then engineered to generate RNA-based sensors functional in the cellular context [75]. The idea of using structural scaffolds from functional riboswitches can expedite the implementation of sensors because these motifs have an efficient switching activity in vivo and, thus, could be better suited for coupling to readout domains. 

SELEX aims at selecting aptamers based on target affinity and selectivity but not on their competence to exert an effective switching activity. The inclusion of an in vivo screening of sensor candidates after SELEX could contribute to overcoming this limitation. Beatrix Suess’ group developed a sensor implementation pipeline that led to the generation of a set of antibiotic sensors in eukaryotic cells. The procedure consists of a few rounds of SELEX, cloning of the resulting library in a reporter gene expression platform, and an in vivo screening for ligand-dependent expression of the reporter system [47,97,98]. On the one hand, the SELEX step enriches the pool in target binders. On the other hand, it reduces the diversity of the initial pool to a library size that allows yeast transformation with high coverage. With a subsequent in vivo screening step, it is possible to fish out only those aptamer candidates that efficiently switch reporter gene expression in response to ligand binding. The recent implementation of Capture-SELEX has tremendously increased the efficiency of this pipeline and represents a key improvement in RNA-based sensor development. Here, the RNA pool is immobilized to a resin containing a sequence tag that hybridizes with a constant region present in all aptamer candidates. An elution step with the free ligand will release only structure-switching aptamers [99]. Alternatively, sensing and actuator domains can be joined by a rational design inspired by the ligand-induced conformational changes of the aptamer. Ortega et al. used SHAPE to examine conformational changes undergone by FBP aptamer upon ligand binding. The aptamer is a stem-loop where FBP binding alters the tertiary structure of the loop without affecting the stem. Thus, the absence of a secondary structure switching activity prompted to design a sensor candidate library based on the HHRz aptazyme framework, which allows engineering RNA devices with ligand-responsive ribozyme tertiary interactions (Figure 1B) [62]. 

Finally, issues related to the sensor readout have also been addressed in the last few years. For instance, light-up sensors yielded low fluorescence due to poor folding and reduced stability of fluorogenic aptamers, especially in mammalian cells. To improve the stability and folding of the sensor construct, Moon et al. embedded it within the three-way junction RNA motif (3WJ), a common fold frequently found in bacterial riboswitches [75]. In specific, the 3WJ motif was used as a scaffold to integrate the sensing and fluorogenic aptamers such that interhelical interactions are only stabilized as a result of ligand binding (Figure 1E) [65]. Litke et al. designed a self-circularizing RNA device framework named *Tornado*. Here, a SAM light-up sensor is inserted between the sequences of two small Twister ribozymes. Upon transcription, Tornado is rapidly self-cleaved in a way that allows a subsequent RNA circularization by a pervasive cellular RNA ligase. The circular RNA sensor is much more stable, achieving a cellular abundance necessary for the accurate detection of metabolite dynamics (Figure 1F) [66]. Another important issue is the robustness of the signal. Natural fluctuations in cellular components and metabolic activity contribute to cell-to-cell heterogeneity (or *extrinsic noise*) within isogenic populations [100]. This noise affects the resolution of sensor readout, such that small signal changes resulting from biologically-relevant metabolite oscillations might become blurred. An additional fluorescence source that does not depend on ligand concentration but is sensitive to extrinsic noise serves to normalize the sensor signal. Thus, the normalization of the fluorescence intensity of the sensor by an unregulated fluorescence source reduces the variability of sensor readout among individual cells [101]. For aptazyme-based sensors of FBP, xanthine, and c-di-GMP in eukaryotic cells, an unregulated fluorescent protein was used to normalize the sensor signal, generating a ratiometric readout (Table 1) [60,62]. In addition, a second fluorogenic aptamer has been recently introduced into a light-up sensor for the first time. Here, a natural aptamer with a 3WJ motif was used as a scaffold to select a new generation of stable fluorogenic aptamers named *Squash*. A yellow Squash was connected to a SAM-sensing aptamer, and this RNA device was integrated with Broccoli in a self-circularizing Tornado framework [72]. This light-up sensor incorporates most design improvements to enhance the stability, folding, and signal emission in a single device and represents a promising tool in metabolite sensor development. 

## 6. Applications of RNA-Based Sensors for Intracellular Metabolites

Sensors for intracellular metabolites have proved their utility in metabolic engineering. The construction of multi-step pathways requires screening enzymatic activities from multiple sources, including enzyme libraries or biological samples from different origins. Quantification of a specific enzyme-product yield is usually performed by analytical methods, such as HPLC, which is time-consuming and only allows for low to medium throughput. Using an RNA-based sensor for this specific product may expedite the identification of a specific enzymatic activity in biological samples or the variant with the highest activity in large enzyme libraries [102]. Michener et al. utilized a well-characterized fluorescent aptazyme-based theophylline sensor to set up an in vivo high throughput screening of enzyme variants in yeast. Here, a caffeine de-methylase library generated by error-prone PCR was transformed into a strain that had the sensor integrated into the genome. A few consecutive cycles of positive and negative selection with FACS led to the identification of an enzyme variant with increased activity and selectivity [103]. Kellenberger et al. utilized the light-up c-di-AMP sensor (Table 1) to screen for the activity of putative diadenylate cyclases co-expressed heterologously in *E. coli*. With this approach, they experimentally validated two predicted diadenylate cyclases from the human pathogen *Clostridium difficile* and the archaea *Methanocaldococcus jannaschii* [77]. 

The enzymes necessary to generate a product of interest are assembled into a synthetic pathway to create a consolidated cell factory, which will still require further optimization. Cell factory optimization entails substantial metabolic engineering, and along this process, the host strain may accumulate mutations, from which only a subset of them contribute to higher product yields, the rest being superfluous or counterproductive. Meyer et al. constructed a whole-cell aptazyme FMN sensor to screen for vitamin B2-secreting cell factories. The authors generated a library of 9000 clones in which a number of DNA segments had been crossed over with wild-type host strain, such that in each clone, a number of the accumulated mutations had been reverted. Individual clones were then co-cultivated with the sensor strain in nanoliter reactor capsules, and B12 producers were selected by FACS [104]. This work benchmarks the utility of RNA-based sensors for the efficient selection of producers after genetic modifications performed during cell factory optimization (e.g., for balancing the expression levels of involved enzymes and modifying cell host metabolism). 

Finally, along the course of industrial fermentations, cell factories typically lose productivity due to the appearance of sub-optimal phenotypes. The fermentation process pursues a high product yield, which is usually in conflict with the selection pressure for an increased growth rate of clones within the cell factory population [105,106]. RNA-based sensors could be used to monitor the appearance of such phenotypic heterogeneity. Torello et al. combined five fluorescent biosensors to monitor the yeast intracellular environment along the fermentation process [107]. These included the aptazyme FBP sensor that reports glycolytic flux (Table 1) [62], along with protein sensors of intracellular pH, ATP levels, oxidative stress, and ribosome production. These biosensors inform about yeast physiology and stress at the single-cell level, providing new means for tracking and controlling the rise of sub-populations. In this regard, the modular architecture of RNA-based sensors that actuate gene expression (e.g., aptazyme) allows coupling a specific functionality (other than fluorescence emission) to a metabolite concentration. Thus, it might be possible to control cell fate in response to a metabolite concentration by replacing the reporter gene. For instance, Yang et al. constructed the *Lysine Riboselector*, where the lysine riboswitch was coupled to the *tetA* gene, a dual selection system used in *E. coli*. The Lysine riboselector was utilized to screen a promoter library of an enzyme crucial for diverting metabolic flux toward lysine biosynthesis. After a few rounds of selection cycles, the optimized L-lysine-producing cell factory was enriched to up to 75% of the total population [108]. The implementation of such ligand-dependent selection systems in cell factories could contribute to maintaining the robustness and extend the homogeneity along the fermentation process. 

## 7. Outlook and Concluding Remarks

Today, the design and functioning of RNA-based sensor frameworks have achieved a higher level of development. Two portable RNA sensor frameworks with different mechanisms of actuation and signal generation have been standardized: aptazyme and light-up-based sensors. In the last few years, an enormous effort has addressed major issues that have been holding back the actual implementation of RNA-based sensors to respond to fundamental questions in biology. However, the number of available sensors for different intracellular metabolites is still very low. In the next few years, it is expected that the developed sensor frameworks will be exploited to scale up the catalog of available sensors.

As sensors for different intracellular metabolites become more widely available, their potential applications will also increase. RNA-based sensors could be applied for monitoring different metabolic flux-signaling metabolites. This will provide a rapid, non-invasive, affordable, and straightforward method to measure metabolic fluxes in real-time and at the single-cell level in live cells, which is not feasible today with alternative techniques. Research towards understanding metabolic heterogeneity in human pathologies as relevant as cancer [109] could highly benefit from the development of RNA-based sensors of flux-signaling metabolites. Indeed, today it is possible to combine different sensors in the same cell [107] and to integrate multiple metabolite inputs in a specific output using a single device [99,110,111,112]. Such devices will provide invaluable multidimensional and systems-level information that will help in understanding cell biology.

## Data Availability

Not applicable.

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
