# Peer review of "Real-Time Assessment of Intracellular Metabolites in Single Cells through RNA-Based Sensors"

_biomolecules, 2023, doi:10.3390/biom13050765_

Round 1

Reviewer 1 Report

This review article offers a comprehensive overview of the natural mechanisms utilized by cells for sensing and regulating metabolite flux, including protein-based networks using FBP as a specific example, and riboswitch-mediated mechanisms as a particular focus. The article then delves into the fundamental design principles of RNA-based sensors, starting from early examples of aptazyme to the recent developments of light-up RNA-based sensors. Additionally, the author examines the current obstacles faced by RNA-based sensor development, including issues related to small molecule-aptamer selection and/or sensor readout, and proposes approaches aimed at overcoming these challenges. Finally, the authors briefly discuss the current and potential applications of these synthetic RNA-based metabolite-sensing tools, providing valuable insights into the field.

In recent years, significant advancements have been made in the field of aptamer-based biosensor development. There are ample opportunities for further tool development, particularly in the area of intracellular metabolite sensing for synthetic biology applications. Therefore, this review article, which summarizes the latest progress in the field, is a timely and valuable contribution. Overall, the review is well-structured, comprehensive, and scientifically solid. While a few minor concerns need to be addressed, I recommend accepting the work for publication.

1.     The first part of the article, which discusses how cells sense intracellular metabolites and transduce them into cellular responses, may be too lengthy, as its connection to the subsequent discussion on riboswitch and RNA-based sensors is somewhat tenuous. While the use of FBP as an example is useful, providing too many details could distract readers from the main point.

2.     Although the abstract and introduction mention that the article will address the current and future applications of synthetic RNA-based sensors of metabolites, the "outlook and concluding remarks" section only briefly touches upon this topic. It would be beneficial for readers if the authors provided one or two examples of current applications of RNA-based sensors. Additionally, a separate section dedicated to the applications of RNA-based sensors could enhance the manuscript.

3.     A typo in Line 141, “metabolites o second messengers”.

Author Response

I would like to honestly thank the reviewer for his/her time and effort reviewing my ms. I fully agree with all the comments and did my best to address the suggestions. Thank to the review the ms has notably improved.

  1. In the revised version of the ms the first section of the paper has been summarized. I removed details and redundant concepts. I think now it is much more straight and fits better with the rest of the ms.
  2. I added a new section "Applications of RNA-based sensors for intracellular metabolites", which ellaborates on recent examples of current applications of RNA sensors in metabolic engineering. I  agree with the reviewer that this section enhances the interest of the ms. Thank you for the suggestion.
  3. Thank you. The typo has been corrected in the revised version of the ms. 

Reviewer 2 Report

In this review, the author begins by discussing how cells in all kingdoms sense intracellular metabolites and transduce them into cellular responses to adapt to environmental changes, highlighting the significance of developing intracellular metabolite sensors. The author then summarizes key findings on bacterial riboswitches and goes on to describe the rationale design, molecular mechanisms, and current progress for synthetic RNA-based sensors. The author further discusses major issues and solutions for the development of RNA-based metabolite sensors and finishes by pointing to current and future applications of them.

Overall, this review is well-written and insightful. The discussion on the use of the three-way junction motif as a commonly identified RNA motif in bacterial riboswitches for RNA-based sensor engineering is particularly intriguing. Moreover, it is worth noting that the development of RNA-based sensors not only enables the detection of intracellular metabolites but also provides valuable insights into which RNA structural motifs are capable of binding to small molecules. This inverse perspective further emphasizes the significance of RNA-based sensors in advancing our understanding of RNA regulatory roles and their potential applications in synthetic biology.

There are a few minor comments I have. Firstly, the content in the last row of Table 1 is the same as the first row. Secondly, it is quite unusual to not reference any literature in the introduction section. Finally, some sentences are quite lengthy, and it would be better to split them up to increase clarity and readability.

Author Response

I would like to honestly thank the reviewer for his/her time and effort reviewing my ms. I fully agree with all the comments and did my best to address the suggestions. I think the revised version of the ms is easier to read.

  1. Thank you. The mistake in Table 1 has been corrected in the revised version of the ms.
  2. I now cite the perttinent literature inn the introduction of the the revised version of the ms.
  3. You are right. I have gone through the whole ms and split long sentences. I also asked a colleague to read the ms, and then I rewrote sentences that might not be clear enough.